# *Mycobacterium abscessus* subsp. *massiliense*: Biofilm Formation, Host Immune Response, and Therapeutic Strategies

**DOI:** 10.3390/microorganisms13020447

**Published:** 2025-02-18

**Authors:** Roseane Lustosa de Santana Lira, Flávio Augusto Barros Nogueira, Rosália de Fátima Penna de Carvalho Campos, Dayenne Regina Mota Ferreira, Pedro Lucas Brito Tromps Roxo, Caio César Santana de Azevedo, Eleonôra Costa Monteiro Gimenes, Ruan Lucas Costa Bastos, Camila Evangelista Carnib Nascimento, Flávia Danyelle Oliveira Nunes, Mayane Cristina Pereira Marques, Carmem Duarte Lima Campos, Camila Guerra Martinez, Adrielle Zagmignan, Luís Cláudio Nascimento Silva, Rachel Melo Ribeiro, Ana Paula Silva de Azevedo dos Santos, Rafael Cardoso Carvalho, Eduardo Martins de Sousa

**Affiliations:** 1Graduate Program in Health Sciences, Federal University of Maranhão—UFMA, São Luís 65080-805, Brazil; roseane.lustosa@discente.ufma.br (R.L.d.S.L.); camila.carnib@ufma.br (C.E.C.N.); flavia.danyelle@ufma.br (F.D.O.N.); marques.mayanne@gmail.com (M.C.P.M.); carmem.campos@discente.ufma.br (C.D.L.C.); melo.rachel@ufma.br (R.M.R.); ana.azevedo@ufma.br (A.P.S.d.A.d.S.); carvalho.rafael@ufma.br (R.C.C.); 2Graduate Program in Biodiversity and Biotechnology, Amazônia—BIONORTE, Federal University of Maranhão—UFMA, São Luís 65080-805, Brazil; augustofabn@gmail.com (F.A.B.N.); adrielle004602@ceuma.com.br (A.Z.); 3Graduate Program in Biosciences Applied to Health, CEUMA University—UNICEUMA, São Luís 65075-120, Brazil; rocam_ped@hotmail.com (R.d.F.P.d.C.C.); dayenneregina@hotmail.com (D.R.M.F.); camila005150@ceuma.com.br (C.G.M.); luiscn.silva@ceuma.br (L.C.N.S.); 4Undergraduate in Medicine, CEUMA University—UNICEUMA, São Luís 65075-120, Brazil; pedrotromps.med@gmail.com (P.L.B.T.R.); caioazevedo1513@gmail.com (C.C.S.d.A.); eleonoragimenes.medicina@gmail.com (E.C.M.G.); lucasruan1500@gmail.com (R.L.C.B.)

**Keywords:** biofilm, *Mycobacterium abscessus* subsp. *massiliense*, immune response

## Abstract

Infection by *Mycobacterium abscessus* subsp. *massiliense* poses a growing public health threat, especially to immunocompromised individuals. The pathogenicity of this mycobacterium is directly linked to its ability to form biofilms, complex structures that confer resistance to antibiotics and the host immune response. The extracellular matrix of the biofilm acts as a physical barrier, hindering the penetration of drugs and the action of the immune system, while also inducing a slow-growth state that reduces susceptibility to antibiotics. Current therapies, which involve prolonged use of multiple antibiotics, are often ineffective and cause significant side effects. Therefore, it is essential to explore new strategies targeting bacterial resistance and biofilm destruction. This narrative review explores the biofilm-forming capacity of *Mycobacterium abscessus* subsp. *massiliense* and the potential of novel therapeutic strategies. Promising approaches include inhibiting biofilm formation, developing drugs with improved penetration of the extracellular matrix, combination therapies with agents that destabilize the biofilm structure, and modulating the host immune response. Investing in research and development of new therapeutic strategies is essential to combat this resistant bacterium and improve patient outcomes.

## 1. Introduction

*Mycobacterium abscessus* subsp. *massiliense* is a rapidly growing non-tuberculous mycobacterium [NTM] that stands out as an opportunistic pathogen, posing an increasing threat to human health [1]. The ubiquity of this organism in environments such as water and soil, combined with its tendency to colonize hospital water systems, heightens the risk of infections, particularly among immunocompromised individuals [2].

*M. abscessus* subsp. *massiliense* has a particular affinity for the respiratory tract [3]. The chronic and progressive nature of these infections leads to a constant decline in pulmonary function, significantly affecting patients’ quality of life and survival [4,5]. *M. abscessus* subsp. *massiliense* causes not only pulmonary infections but also skin and soft tissue infections, particularly following surgical procedures, trauma, or the use of medical devices [6,7]. These infections are persistent and often recurrent, posing significant challenges for clinical management [1,7].

One of the challenges in treating infections caused by *M. abscessus* subsp. *massiliense* is its remarkable resistance to a wide range of antibiotics [8]. This intrinsic resistance results from a combination of factors, including the complex structure of the mycobacterial cell wall, the presence of efflux pumps that actively expel drugs, and its ability to form biofilms, which provide protection against environmental stress and antibiotic action [7,9].

Biofilms are communities of microorganisms that adhere to surfaces and produce a self-generated extracellular matrix. This matrix provides antimicrobial resistance and helps microorganisms evade the host immune response [10,11]. The extracellular matrix, primarily composed of polysaccharides, proteins, and extracellular DNA, acts as a physical barrier, limiting the penetration of antibiotics. Simultaneously, the slow growth and reduced metabolism of bacteria within the biofilm make them less susceptible to antibiotics [12].

Furthermore, biofilm formation can promote bacteria within the biofilm to express antibiotic resistance genes and harbor persistent cells, which are highly tolerant to antibiotics and can cause relapses after treatment [10]. The extracellular matrix also shields bacteria from the immune system, while intercellular communication through *quorum sensing* coordinates biofilm formation, virulence, and antibiotic resistance [10,11,13].

It is essential to note that biofilm formation is not restricted to a single morphotype of *M. abscessus*. Both smooth and rough morphotypes have demonstrated the ability to form biofilms [12]. Rough morphotypes, often associated with higher virulence, form biofilm-like aggregates that exhibit tolerance to adverse conditions such as acidic pH, hydrogen peroxide, and antibiotic treatments [10]. Cord formation, a hallmark of rough strains, facilitates extracellular dissemination and immune evasion, contributing to infection persistence [8,10,11].

Biofilm formation by *M. abscessus*, particularly the *massiliense* subspecies, presents a significant obstacle in the treatment of pulmonary infections, especially in patients with cystic fibrosis [5]. The biofilm’s complex structure, composed of bacteria embedded in an extracellular matrix, creates a protective environment that limits antibiotic penetration and hinders the host immune system’s response [11,12]. The increasing prevalence of *M. abscessus* infections and its inherent antimicrobial resistance demand a deeper understanding of the mechanisms behind biofilm formation, composition, and structure [5,8]. Elucidating the intricate relationship between biofilms and the host immune response, including immune evasion mechanisms and infection persistence, is critical for developing effective therapeutic strategies.

This review aims to investigate the impact of biofilm formation by *M. abscessus* subsp. *massiliense* on the host immune response and explore new therapeutic approaches, including the development of novel antimicrobial agents, therapies targeting the biofilm matrix, and strategies to enhance the activity of existing antibiotics, with the goal of improving clinical outcomes for patients affected by this challenging bacterium.

## 2. Characteristics of *M. abscessus* subsp. *massiliense*

*M. abscessus*, including the subspecies *massiliense*, is a rapidly growing mycobacterium characterized by its ability to form visible colonies on agar media within just four days [14]. This species exhibits two distinct colony morphologies: smooth [MaSm] and rough [MaRg], differentiated by the presence or absence of glycopeptidolipids [GPLs] in the cell wall. The MaSm variant, rich in GPLs, presents a smooth surface and a nearly mucoid appearance when forming biofilms [12]. In contrast, the MaRg variant, deficient in GPLs, is hyperaggregative, forming biofilm-like aggregates with a characteristic cauliflower morphology [8]. Figure 1 illustrates the morphological variations between MaSm and MaRg colonies and also details the cell wall structure and the distribution of GPLs, which are important for the bacterium’s ability to cause disease, including components such as GPLs and LAM.

Under microscopy, *M. abscessus* subsp. *massiliense* appears as short or long, thin, slightly curved, non-sporulated, acid-fast bacilli, staining red with the Ziehl–Neelsen technique. Genomically, *M. abscessus* subsp. *massiliense* is recognized as a distinct subspecies of *M. abscessus* through genomic analyses [11]. A key genetic feature of the *massiliense* subspecies is the presence of an inactivated erm41 gene, which confers susceptibility to macrolide antibiotics [15].

Additionally, the genome of *M. abscessus* harbors non-mycobacterial virulence genes, some originating from *P. aeruginosa* [16]. Mutations in genes involved in the biosynthesis or transport of GPLs can lead to transitions from high-GPL-producing to low-GPL-producing variants, affecting virulence and interactions with host cells [8]. Similarly, modifications in lipoarabinomannan [LAM], an important lipoglycan in the cell wall, due to mutations in the *embC* gene, contribute to phenotypic heterogeneity in host-adapted *M. abscessus* isolates. These mutations impact the ability to form serpentine cords and biofilms, replicate within innate immune cells, and induce inflammatory responses [17].

The intricate architecture of the *M. abscessus* cell wall, rich in unique components such as GPLs, LAM, mycolic acids, and glycolipids, grants this mycobacterium remarkable adaptability and survival in diverse environments. This complex structure, along with antibiotic resistance mechanisms mediated by efflux pumps and the ESX-4 secretion system, significantly contributes to its pathogenicity, enabling the bacterium to persist in intracellular environments, modulate the host immune response, and cause hard-to-treat infections [11,15,16].

## 3. Virulence Factors

The virulence of *M. abscessus* subsp. *massiliense* is multifactorial and intrinsically linked to its ability to form biofilms—complex structures that confer antibiotic resistance and enable immune evasion. Several virulence factors influence infection and antibiotic resistance, with the key factors presented in Table 1 [14].

The morphotypes are among the factors that directly influence the virulence of this bacterium. The smooth morphotype (MaSm), rich in GPLs, is associated with the formation of more viscous biofilms and immune system evasion, while the rough morphotype (MaRg), with fewer GPLs, is more invasive and forms aggregates resistant to adverse conditions. The transition between these morphotypes reflects the bacterium’s adaptation to the host environment, and the mechanisms of immune evasion are depicted in Figure 2 [14,23].

GPLs are key components of the cell wall, influencing biofilm formation and modulating the immune response. They inhibit the activation of immune system receptors, such as TLR2, and the production of pro-inflammatory cytokines like TNF-α, favoring infection persistence [3,5]. Another important factor is trehalose dimycolate (TDM), a glycolipid present in both morphotypes, contributing to virulence, especially in biofilms formed in the pulmonary environment of cystic fibrosis patients. In addition to GPLs and TDM, lipoarabinomannan [LAM] and surface proteins also play a role in the bacterium’s pathogenicity [6,8,19].

Biofilm formation begins with the adhesion of planktonic bacteria to a surface, followed by the production of the extracellular matrix (ECM), composed of polysaccharides, proteins, eDNA, and lipids [13,14]. This matrix provides antibiotic resistance and protects the bacteria from the immune system. Bacterial communication, known as quorum sensing, and environmental factors such as the presence of nutrients, ions, and temperature influence biofilm formation and development. The biofilm progresses through different stages, with changes in the expression of genes related to metabolism and virulence [5,10,11].

The virulence of *M. abscessus* is a complex process involving the interaction of various factors, including biofilm formation, morphological variation, the production of virulence factors, and adaptation to the host environment. Understanding these mechanisms is essential for developing new therapeutic strategies that overcome resistance and eradicate the infection [20,21].

## 4. Biofilm Formation

The formation of biofilm by *M. abscessus* occurs in three main stages: first, adhesion, where planktonic cells attach to a surface; then aggregation and maturation, where bacteria cluster to form microcolonies and produce the ECM, maturing the biofilm [24].

The adhesion of planktonic cells to a surface is facilitated by GPLs and influenced by the hydrophobicity of the surface [24,25]. The bacteria aggregate, forming microcolonies and producing the ECM, which is composed of polysaccharides, proteins, eDNA, and lipids. This matrix acts as a protective barrier [26]. The ECM provides antibiotic resistance and impedes the immune system’s action, with its composition varying to influence the biofilm’s characteristics [26,27,28].

As the biofilm matures, it becomes more structured, with channels that allow nutrient circulation and waste removal [26,27]. Bacterial communication, known as quorum sensing, regulates gene expression and coordinates the behavior of the bacterial community [29]. The mature biofilm reaches its maximum resistance to antimicrobials [28]. In response to environmental factors or internal signals, the biofilm may disperse, releasing bacterial cells that can colonize new areas, as shown in Figure 3 [30].

Environmental factors, such as nutrient availability, oxygen presence, and pH, also influence biofilm formation [3,5,20]. To combat this resistance, various strategies can be employed, such as inhibiting bacterial adhesion, modulating GPL production, inhibiting quorum sensing, and blocking efflux pumps [31,32,33]. The destabilization of the ECM can be achieved with enzymes that degrade its components, such as DNases and enzymes that break down polysaccharides, or with disaggregating agents like Tween 80 [13,30,34].

Other strategies include increasing antibiotic penetration through the development of new drugs or the use of nanoparticles and modulating the immune response with immunotherapies or cytokines [31,32,33]. The use of liposomal antibiotics, such as liposomal amikacin and liposomal ciprofloxacin, has shown potential to penetrate biofilms and increase treatment efficacy [14,25,34]. Antimicrobial peptides can increase the sensitivity of *M. abscessus* to antibiotics in mature biofilms [34], and disruption of bacterial aggregates, combined with immune response modulation, can also improve the response to antibiotics [14,30,31,34].

The treatment of *M. abscessus* infections requires a multidisciplinary approach, combining different strategies, such as biofilm matrix destruction, the development of new antibiotic molecules, immunomodulation, and phage therapy [3,5]. Combined therapy, which associates different strategies like using antibiotics with agents that destabilize the biofilm or phage therapy, may be more effective than using each therapy in isolation. Other approaches include the use of agents that increase oxygen concentration in the environment and beta-lactamase inhibitors [23,24,25,32].

Treatment strategies to combat *M. abscessus* should target immune evasion mechanisms. Biofilm disintegration, phage therapy, and the development of new antibiotics with better activity, especially in biofilms, are promising approaches. To overcome bacterial resistance, combining drugs with different mechanisms of action is essential. Furthermore, modulating the host’s immune response is crucial to controlling infection by this mycobacterium [3,5].

## 5. Immune Response

The innate immune response to *M. abscessus* subsp. *massiliense* serves as the body’s first line of defense. Cells such as macrophages and neutrophils act as phagocytes, engulfing and destroying the bacteria [11,14,15]. Macrophages can be infected by both the smooth and rough variants of *M. abscessus*, although the rough variant survives longer within these cells. Neutrophils are also critical for defense against *M. abscessus*, but the bacterium can induce a limited activation pattern in neutrophils, which promotes its survival [8]. Natural killer (NK) cells play a role by destroying infected cells, contributing to infection control [5,14].

Biofilms formed by *M. abscessus* subsp. *massiliense* pose a significant challenge to the innate immune response [13]. The extracellular matrix of the biofilm acts as a physical barrier, hindering the access of immune cells and impairing phagocytosis. Additionally, bacterial aggregation within biofilms increases tolerance to antibiotics, making infection eradication more difficult [4,5].

When the innate immune response fails to control the infection, the adaptive immune response is activated [5]. CD4+ T cells play a pivotal role in the immune response against mycobacteria, particularly in patients with cystic fibrosis. These cells produce cytokines such as interferon-gamma (IFN-γ) and tumor necrosis factor-alpha (TNF-α). IFN-γ activates macrophages, enhancing their bactericidal capacity, while TNF-α contributes to granuloma formation, which aims to contain the infection [4,5,18].

The Cytokines TNF and IFN-γ produced by macrophages and CD4+ T cells are required for granuloma formation and induce macrophage effector function, such as phagosomal acidification and reactive oxygen species (ROS) production, while Type I IFN production promotes nitric oxide (NO) production. IL-8 attracts neutrophils, which mediate phagocytosis, NET production, and antimicrobial peptide LL37 secretion.

However, the release of *M. abscessus* into the extracellular space as a result of cell death leads to the formation of serpentine cords, which are resistant to innate immune defenses and results in uncontrolled bacterial growth [35,36]. *M. abscessus* possesses numerous immune evasion mechanisms to resist macrophage effector functions. These include: bacterial escape from the phagosome to the cytosol by interfering with the phagosomal membrane; prolonged survival within the phagosome by blocking phagosomal acidification and thus preventing degradation; and inhibition of macrophage TLR signaling, which limits downstream immune cell activation and recruitment [5,35,36]. *M. abscessus* also persists extracellularly by avoiding phagocytosis. This is achieved through adherence to macrophage phagocytic cups and by forming serpentine cords too large for macrophage engulfment [5,35,36,37,38].

The production of IgG antibodies against *M. abscessus* is also observed in patients with cystic fibrosis. These antibodies may serve as biomarkers of infection, although their protective role against *M. abscessus* remains unclear [15,18]. The biofilm’s nature likely limits the effectiveness of these antibodies. Research has demonstrated that infection by *Mycobacterium tuberculosis* and the Bacillus Calmette–Guérin (BCG0 vaccine can induce a cross-reactive memory lymphocyte response against *M. abscessus* and the *Mycobacterium avium* complex [15,39].

However, despite the complex interaction between *M. abscessus* subsp. *massiliense* and the host’s immune system, the immune response is often insufficient to control the infection. This difficulty in eradicating the infection highlights the need to develop new therapeutic strategies that complement the host’s immune response and assist in the elimination of the bacteria [8,11,14,18].

## 6. Therapeutic Strategies

The difficulty in eradicating *M. abscessus* infections due to the complex interaction between the bacteria and the immune system, as well as the bacterium’s intrinsic resistance to antibiotics, primarily due to its ability to form biofilms, makes it essential to develop new therapeutic strategies that can circumvent these evasion and resistance mechanisms of the bacteria [40,41].

The innate resistance of *M. abscessus* subsp. *massiliense* is attributed to several factors, including its complex cell wall structure, efficient efflux pump systems, and the production of antibiotic-modifying enzymes [17]. The mycobacterial cell wall, rich in lipids, acts as a formidable barrier, restricting the penetration of many antibiotics. Efflux pumps, specialized membrane proteins, actively expel antibiotics that manage to penetrate the cell, reducing their intracellular concentration and effectiveness. Additionally, the bacterium produces enzymes that chemically modify antibiotics, neutralizing their activity [13,14].

The ability of *M. abscessus* to form biofilms compromises the effectiveness of treatment for this infection, requiring strategies aimed at breaking this protective barrier and enhancing the efficacy of antibiotics [19,37,38]. Several approaches show promise in this regard. Enzymes that degrade the components of the extracellular matrix, such as polysaccharides, proteins, and DNA, can destabilize the biofilm, exposing the bacteria to antimicrobials. Detergents, like Tween 80, have also shown the ability to disrupt biofilms, facilitating the action of antibiotics [24,29,30].

Another innovative targeted strategy is the use of iron chelators, which aim to sequester iron and modulate the environment, limiting the growth of the mycobacteria [41,42,43]. Genetic editing and nanotechnology are examples of advanced therapeutic approaches. Genetic editing can be used to silence genes that promote biofilm formation, while nanotechnology delivers drugs directly to the site of infection [2,44].

The development of new drugs with greater penetration ability into the extracellular matrix is essential for effective treatment of *M. abscessus* subsp. *massiliense* infections [4,6,45]. These drugs should have characteristics that favor their diffusion through the matrix, such as higher liposolubility and lower molecular weight, enabling them to reach the bacteria protected within the biofilm [46].

Combined therapy, using agents that destabilize the biofilm structure alongside traditional antibiotics, emerges as a promising approach to eradicate these infections. By combining different mechanisms of action, this strategy aims to overcome the challenges posed by biofilms and enhance treatment efficacy [45,46,47].

The standard treatment for pulmonary infections caused by *M. abscessus* subsp. *massiliense* involves a prolonged regimen of multiple antibiotics, lasting 12 to 24 months [10,11,45,47]. Typical antibiotic combinations include macrolides [clarithromycin or azithromycin], aminoglycosides [amikacin], and other agents such as rifampicin, ethambutol, imipenem, cefoxitin, and linezolid. The specific choice of antibiotics is guided by susceptibility testing of the isolate and the patient’s clinical condition. However, these therapies often yield limited efficacy, with low success rates, particularly in biofilm-associated infections [5,18].

Given the challenges posed by antibiotic resistance and biofilm formation, new therapeutic strategies are being explored. Efforts to identify new antibiotics with enhanced activity against *M. abscessus* subsp. *massiliense*, especially those capable of penetrating biofilms, are underway [10]. NF1001, a novel thiopeptide antibiotic, has demonstrated potent activity against both planktonic forms and biofilms of *M. abscessus* subsp. *massiliense*. Additionally, gold-complexed sulfadiazine has shown promise in reducing biofilm formation by inhibiting c-di-GMP synthesis, a signaling molecule essential for biofilm development [10,13,17].

Monoclonal antibodies targeting essential biofilm components, such as DNABII proteins, can destabilize biofilm structure, exposing bacteria to antibiotics and the host immune system. Studies with the humanized monoclonal antibody HuTip*Mab* have shown promising results, enhancing the susceptibility of *M. abscessus* subsp. *massiliense* to amikacin and azithromycin [11].

Disrupting bacterial communication systems, known as quorum sensing, can inhibit biofilm formation or make it more susceptible to antibiotics. While this approach shows promise for controlling biofilm-related infections, it remains in early development [13,17]. Similarly, enzymes that degrade the extracellular matrix of biofilms can dismantle their structure, rendering bacteria more vulnerable to antibiotics and the host immune response [10]. Modulating the host immune response to enhance T-cell activity and pro-inflammatory cytokine production may also improve the body’s ability to combat *M. abscessus* subsp. *massiliense* [11].

Finally, bacteriophages—viruses that infect and destroy bacteria—emerge as a promising alternative to antibiotics, particularly for treating infections caused by multidrug-resistant strains [5]. The infections caused by *M. abscessus* subsp. *massiliense* are significant therapeutic challenges. The bacterium’s intrinsic resistance to multiple antibiotics, its ability to form biofilms, and its adaptation to the hostile pulmonary environment contribute to the difficulty in treating these infections.

### 6.1. Conventional Approaches

The standard treatment for pulmonary infections caused by *M. abscessus* subsp. *massiliense* typically involves a combination of intravenous and oral antibiotics administered over an extended duration, ranging from months to years. Antibiotic selection is guided by in vitro susceptibility testing, though the correlation between these test results and clinical response is often weak [47,48]. A summary of commonly used antibiotics is presented in Table 2.

The selection of antibiotics to treat infections caused by mycobacteria should always be based on in vitro susceptibility tests, which indicate which drugs are most effective against the specific bacterium [22]. Treatment is generally long term, lasting months or even years, and the combination of different antibiotics is essential to prevent the bacteria from developing resistance to the medications. It is critical to monitor the side effects of antibiotics, especially during prolonged treatments [46,49].

The guidelines from the American Thoracic Society/Infectious Diseases Society of America (ATS/IDSA) in 2007 and the British Thoracic Society (BTS) in 2017 recommend a minimum of 12 months of treatment following culture conversion. This means that treatment should continue for at least one year after sputum culture test results become negative [11].

Traditional antibiotic treatment for *M. abscessus* subsp. *massiliense* infections faces several challenges [41]. Success rates are low, ranging from 10% to 55%, and the bacteria can persist in the lungs even after treatment [3,22,23]. Furthermore, prolonged use of high-dose antibiotics causes side effects such as bone marrow problems, liver issues, and allergic reactions [2,21]. While the host immune response is essential for controlling infections, it is often ineffective against bacteria within biofilms. These biofilms shield bacteria from immune cells and antimicrobial agents, making infections difficult to eradicate [24].

### 6.2. Emerging Therapeutic Strategies

The search for new therapeutic strategies for *M. abscessus* subsp. *massiliense* infections has intensified, driven by the limitations of conventional antibiotic therapy. This urgent need for therapeutic alternatives is highlighted by the analysis of strategies reported annually between 2000 and 2025, as shown in Figure 4.

The data, compiled from searches in databases such as Scopus, Web of Science (WOS), CINAHL, EMBASE, WHO, and Pubmed/Medline, reveal a significant increase in the number of strategies reported in 2024, contrasting with the absence of records in certain years between 2000, 2001, and 2003 to 2010. This temporal disparity highlights the growing scientific interest in developing new approaches to combat this challenging pathogen, emphasizing the importance of ongoing research to expand the therapeutic arsenal against *M. abscessus* subsp. *massiliense* infections.

Research into new therapeutic strategies, such as new antibiotics, antiparasitic agents, phage therapy, immunotherapy, biofilm-targeted approaches, enzymes, gene editing and nanotechnology (Table 3), is essential to improving the prognosis of patients infected with this bacterium [5,20,31]. Combining different therapeutic approaches, such as antibiotics with phage therapy, may help overcome drug resistance and persistent infections, which are major challenges in treating *M. abscessus* [6,23].

Bedaquiline and tedizolid emerge as alternatives in the treatment of *M. abscessus* infections [12,14,17,34]. Initially approved for multidrug-resistant tuberculosis, bedaquiline inhibits bacterial energy production, proving effective against *M. abscessus*, including resistant strains. Studies indicate that its combination with amikacin and rifabutin may be bactericidal, acting on dormant forms and biofilms. However, bedaquiline has limitations in inhibiting biofilm formation, particularly in rough morphotypes [17,34].

Tedizolid, an antibiotic in the oxazolidinone class, inhibits bacterial protein synthesis. Although it exhibits in vitro activity against *M. abscessus*, its clinical efficacy still requires further study. Data suggest that its monotherapy may be insufficient to eradicate the bacterium [17,34].

Both drugs are frequently used in combination with other antibiotics to enhance efficacy and combat resistance. The use of drugs already approved for other indications, such as bedaquiline and tedizolid, accelerates the discovery of new therapies [10,12,14,17,34]. Given the variability in response among *M. abscessus* subsp. *massiliense*, a personalized approach based on susceptibility testing is necessary [33,34,35].

NF1001, a novel thiol peptide antibiotic derived from an Antarctic soil sample, demonstrates a broad spectrum of activity against NTM, including *M. abscessus* (Mabs). It acts against both planktonic forms and biofilms, including multi-drug-resistant strains [34]. NF1001 exhibits concentration-dependent activity, with minimum inhibitory concentrations (MICs) ranging from 0.06 to 2 µg/mL for various NTM species [8,10,14,34].

In Mabs biofilms, NF1001 reduces bacterial load, with results comparable to azithromycin. Its activity in biofilms is similar to amikacin. NF1001 also demonstrates intracellular efficacy against NTMs in macrophages. Synergy studies show that NF1001 works in combination with commonly used NTM antibiotics, such as amikacin, clarithromycin, and moxifloxacin [34]. Its mechanism of action appears distinct from traditional protein synthesis inhibitors, possibly interfering with ribosomal protein synthesis. Toxicity studies indicate a promising safety profile for NF1001, which also shows good intracellular penetration [8,10,12,14,17].

Another therapeutic approach that demonstrates biofilm elimination capability is bithionol, an antiparasitic drug that emerges as a promising alternative [8,12,14,17]. Studies show its potent antimicrobial activity against *M. abscessus*, with minimum inhibitory concentrations (MICs)ranging from 0.625 µM to 2.5 µM and proven bactericidal action.

Bithionol exhibits a remarkable ability to eliminate *M. abscessus* biofilms, eradicating bacteria at concentrations of 128 µg/mL [50,51,54]. This property distinguishes it from conventional antibiotics, such as clarithromycin, amikacin, and moxifloxacin, which show low efficacy against biofilms. The mechanism of action of bithionol involves changes in cell morphology and bacterial lysis [11,35,48].

Phage therapy emerges as a promising alternative for treating *M. abscessus* infections. Phages, viruses that infect bacteria, offer high specificity to *M. abscessus*, minimizing damage to the patient’s microbiota [10,51,55]. These viruses destroy bacteria through cell lysis, acting even in biofilms, where conventional antibiotics struggle to penetrate. Additionally, they exhibit immunomodulatory potential, beneficial in pulmonary diseases [10].

Despite its potential, phage therapy faces challenges such as bacterial resistance development and interference from temperate phages that integrate into the host bacterium’s genome [56]. To overcome these obstacles, researchers are exploring the use of phage cocktails with different mechanisms of action. Administration may be performed via inhalation for pulmonary infections, intravenously for systemic infections, and topically for skin infections. Encapsulation in liposomes enhances therapy effectiveness [56,57].

The combination of phages with antibiotics shows promise, creating synergy and facilitating drug action. Success cases with genetically modified phages illustrate the potential of this therapy [40,42]. While still in its early stages, preclinical research using animal models and reports of compassionate use indicate positive results. The search for new phages, genomic analysis, and in vitro and in vivo studies are crucial for advancing phage therapy. Combined with other strategies, phage therapy appears to be an option in the treatment of *M. abscessus* infections, particularly in cases of multidrug resistance [42,43,44,58,59].

Another significant therapeutic approach is host-directed immunotherapy, which emerges as a promising strategy to modulate the immune system and combat infection [1,13]. This approach aims to strengthen the body’s natural defenses, complementing or replacing traditional antimicrobial therapies [51,52].

One of the main challenges in *M. abscessus* infections is inadequate immune response, exacerbated by biofilm formation that protects the bacteria [5,10,41,45]. Dysregulated inflammatory responses can cause damage to lung tissues [47]. Immunotherapy aims to activate host defense mechanisms to eliminate the bacterium, including in biofilms. This can be achieved by activating antioxidant pathways, modulating immune cell activity such as neutrophils and macrophages, and controlling inflammatory cytokine production [45,46,47].

Biofilm disruption is essential for the success of immunotherapy, as it facilitates the action of immune cells and antibiotics [5,45,47]. Studies with immune checkpoint inhibitors and activators of the antioxidant NQO1 pathway show promising results. Human airway organoid models and co-cultures of *M. abscessus* with immune cells allow for the investigation of host-pathogen interactions and the evaluation of new therapies [11].

GPLs on the surface of *M. abscessus* mask the underlying phosphatidylmyo-inositol mannosides in the cell wall, preventing interaction with TLR2 (Toll-like receptor 2) and consequently inhibiting TNF-α induction in macrophages. Spontaneous loss of GPLs in rough variants may lead to a more intense immune response, with increased inflammation [48,50,54].

The structure of lipoarabinomannan (LAM), a lipoglycan present in the cell wall, also influences the interaction with TLR2 and the inflammatory response [54]. Mutations in the embC gene, which codes for an arabinosyl transferase involved in LAM synthesis, may lead to changes in LAM structure and modify TLR2 activation, as well as affect cord and biofilm formation [48,50,51,54].

Although *M. abscessus* can induce the production of pro-inflammatory cytokines such as TNF-α and IL-6, the inflammatory response can be regulated to prevent pathogen elimination and, in some cases, promote infection persistence [4,5,18,48]. The ability to form cords [cord-like structures] and biofilms is associated with evasion of phagocytosis, promoting extracellular growth and infection persistence [48,50,54].

*M. abscessus* uses a combination of mechanisms to evade the immune response, including biofilm formation, modulation of the inflammatory response, and the ability to survive inside macrophages [52]. Combining immunotherapy with antibiotics and biofilm-disrupting therapies enhances the antibacterial response. Despite progress, immunotherapy for *M. abscessus* is still under development. Further research is needed to identify optimal therapeutic targets and understand the bacterium’s immune evasion mechanisms [10,52,57].

Another promising approach is avibactam, a β-lactamase inhibitor, which serves as a solution to restore the efficacy of antibiotics [23,24,25,27]. The β-lactamase BlaMab, produced by *M. abscessus*, inactivates β-lactam antibiotics, rendering them ineffective. Avibactam inhibits BlaMab, allowing antibiotics to regain activity against the bacterium. In vitro and in vivo studies demonstrate that combining avibactam with imipenem significantly enhances antibacterial activity against *M. abscessus* [56,57]. In animal models, this combination reduces bacterial load and improves treatment outcomes [57].

Resistance to β-lactams is a challenge in treating *M. abscessus* infections, and avibactam offers a new strategy to overcome this problem [27]. The combination of avibactam with imipenem shows promise, with potential to improve the treatment of these infections. It is important to note that avibactam has no intrinsic antimicrobial activity, acting specifically as an inhibitor of BlaMab [23,24,25,27].

Although the results are promising, more research is needed to determine the best clinical use of avibactam, including dosage optimization and combinations with other antibiotics [17,34]. Avibactam represents a significant advancement in treating *M. abscessus* infections, restoring the activity of β-lactam antibiotics and paving the way for new therapeutic strategies [25,55].

Therapeutic approaches targeting biofilms are also promising strategies in treating *M. abscessus* infections. These approaches focus on disrupting the biofilm structure and increasing bacterial susceptibility to antimicrobial agents. Three promising approaches are the use of the detergent Tween 80, increased oxygenation, and iron chelation [2,3,17,29,60].

Tween 80 works by disaggregating *M. abscessus* biofilms, breaking down the extracellular matrix and facilitating antibiotic access to the bacteria [13]. Furthermore, Tween 80 enhances aerobic respiration and bacterial growth, making them more susceptible to antibiotics that target metabolically active bacteria [29,40].

Oxygenation also plays a crucial role in combating *M. abscessus*. Under low oxygen conditions, the bacterium enters a dormant state, reducing its susceptibility to antibiotics [8,10,14]. Increasing oxygen levels stimulates bacterial metabolism and enhances the action of antibiotics such as amikacin [42,59].

Combining Tween 80 and oxygenation may have a synergistic effect in treatment. Biofilm disaggregation by Tween 80 allows oxygen to penetrate deeper layers of the structure, increasing antibiotic efficacy [17,34]. This combination represents a promising strategy for developing more effective therapies against *M. abscessus* infections [60,61].

Iron chelation therapy has emerged as an innovative approach in the fight against bacterial infections, particularly against multidrug-resistant bacteria such as *M. abscessus* [6,41,42,43]. This strategy works by sequestering iron, a critical element for bacterial metabolism, indirectly modulating the host environment to limit bacterial growth and enhance susceptibility to antibiotics [41].

Iron chelators, such as deferoxamine, bind to available iron in the body, making it inaccessible to bacteria [42,43]. This iron deprivation disrupts various bacterial processes, including biofilm formation—structures that contribute to antibiotic resistance. By interfering with biofilm formation, iron chelators potentiate the action of antibiotics and facilitate infection eradication [41,42,43].

Iron chelation therapy offers several advantages. Since it targets the host rather than the bacteria, it reduces the risk of resistance development [41,42]. Additionally, it can modulate the immune response, reducing inflammation and promoting bacterial clearance [43]. It is especially effective against persistent bacteria, which enter a dormant state and become tolerant to conventional antibiotics [60,61].

Combining iron chelators with traditional antibiotics, biofilm-disrupting agents, and other host-directed therapies, such as immunomodulators, can increase treatment efficacy [41]. However, challenges remain, including the potential for resistance to chelators, host toxicity, and the drug’s bioavailability at the site of infection [60,61].

The ECM of *M. abscessus* biofilms acts as a protective barrier, hindering antibiotic penetration and immune system action [9,10,55,61]. Enzymes capable of degrading the ECM present a promising alternative for treating these persistent infections [9].

Studies show that various enzymes target specific ECM components [52]. Phospholipases, for example, promote biofilm dispersion by degrading phospholipids in the matrix [46]. Carbohydrases, such as α-mannosidase and cellulase, break down specific carbohydrates, while proteases, such as proteinase K, degrade proteins in the ECM [19,31,60]. The activity of these enzymes depends on the culture medium and environmental conditions, suggesting that the ECM’s composition is influenced by these factors [6,7].

The ECM of *M. abscessus* consists of lipids, carbohydrates, proteins, and extracellular DNA (eDNA) [5,7,47]. The proportion of each component varies, with lipids being the most abundant, followed by eDNA. This complex and variable composition explains the differential response to enzymatic treatments [47].

The degradation of the ECM by enzymes can increase the susceptibility of *M. abscessus* to antibiotics, facilitating their penetration and action. Combining enzymes with antibiotics or other therapeutic agents, such as DNase to degrade extracellular DNA (eDNA0, can potentiate treatment effectiveness [47,48,50]. Moreover, the enzyme-induced biofilm dispersion may enhance the efficacy of other therapies, such as oxygenation, which boosts bacterial metabolism and antibiotic susceptibility [3,34].

It is important to note that the effectiveness of enzymes varies depending on environmental conditions and the maturation stage of the biofilm. Additionally, bacteria may develop resistance to these treatments [39,45,61]. The development of new approaches and the combination of different strategies are essential for combating persistent *M. abscessus* infections [19,40]. Enzymatic degradation of the ECM is a promising strategy for treating *M. abscessus* infections. Identifying specific enzymes and combining them with other therapies may be key to overcoming bacterial resistance and improving clinical outcomes [2,41,46].

Genetic editing allows for silencing genes essential for biofilm formation and bacterial virulence. The Tet-OFF system, for example, controls the expression of specific genes, paving the way for the development of new therapies [57]. Studies involving the mmpL3 gene, which is involved in cell wall biosynthesis and the transport of TDM, have shown that its inactivation reduces biofilm formation and bacterial virulence [58].

Genetic editing can also be used to silence antibiotic resistance genes and genes involved in the synthesis of GPLs, which are critical for adhesion and biofilm formation [58]. Furthermore, this technology helps investigate resistance mechanisms, aiding in the search for new therapeutic strategies [39,45].

Nanotechnology offers tools for targeted drug delivery, such as nanoparticles that encapsulate antibiotics and facilitate their penetration into the biofilm’s extracellular matrix [3,4]. Liposomes containing antibiotics have proven effective in reducing bacterial load in biofilms, and inhalation of liposomal amikacin improves drug penetration into the biofilm [47,59].

Nanoparticles can also be designed to destabilize biofilm structures, targeting specific ECM components like eDNA [47,49,53]. Additionally, nanotechnology enables the development of new antimicrobial materials, such as silver nanoparticles, which can be incorporated into medical devices to prevent biofilm formation. Nanomaterials may also be used in sensors to monitor infection progression and treatment efficacy [53].

The combination of genetic editing with nanotechnology expands therapeutic possibilities [2,54]. Genetic editing can silence genes that promote biofilm formation, while nanotechnology delivers drugs directly to the site of infection. This combined approach, along with genetic data from the patient, paves the way for personalized therapies that are more effective and have fewer side effects [2,54,61].

The pursuit of new therapeutic strategies to combat *M. abscessus* is crucial to overcoming the challenges posed by bacterial resistance and improving patient prognosis. Continued investment in research and innovation in new therapies is essential for advancing the treatment of this challenging infection.

## 7. Conclusions and Future Perspectives

*M. abscessus* subsp. *massiliense* infections present a growing challenge to public health, particularly for immunocompromised individuals. The bacterium’s ability to form biofilms, complex structures that house bacterial communities, is crucial for its pathogenicity and resistance to treatment.

Biofilms act as protective barriers against antibiotics and the host’s immune response, making infections caused by *M. abscessus* subsp. *massiliense* notoriously difficult to treat. The complex biofilm structure, the bacterium’s intrinsic resistance to antibiotics, and its ability to adapt to the hostile environment of the host’s lungs contribute to the difficulty of eradicating the infection. Biofilm formation is influenced by various factors, including the presence of lipoglycopeptides (LGPs) in the bacterial cell wall, the quorum sensing communication system, and environmental conditions such as nutrient and oxygen availability. Although the host’s immune response is crucial for controlling infections, it is often ineffective against bacteria within biofilms. These biofilms shield bacteria from immune cells and antimicrobial agents, making infection eradication challenging.

The combination of different approaches, such as the use of antibiotics, appears promising in overcoming drug resistance and infection persistence. Continued research and the development of new therapeutic strategies for treating *M. abscessus* subsp. *massiliense* infections are essential, including the investigation of novel combination therapies, host-directed therapies, and effective immunotherapies.

## Figures and Tables

**Figure 1 microorganisms-13-00447-f001:**
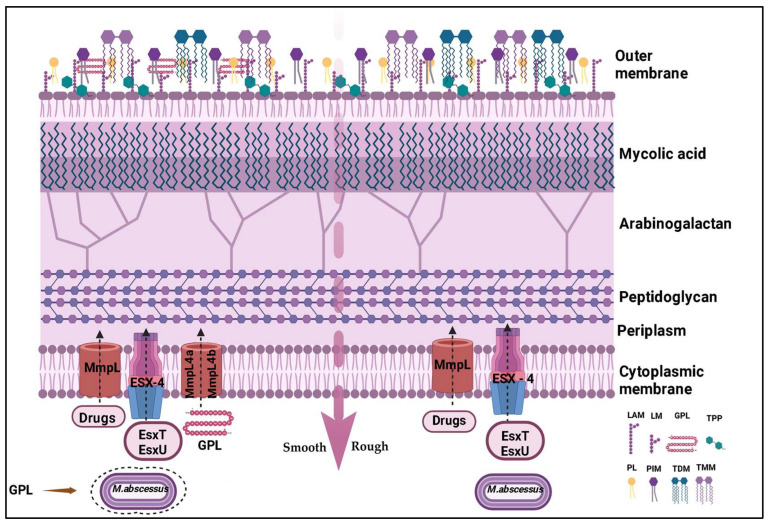
Schematic representation of the mycobacterial cell envelope. GPLs are specific to the smooth [S] variant cell wall and are transported across the inner membrane by MmpL4a and MmpL4b proteins. Other MmpL proteins may function as drug efflux pumps, contributing to intrinsic antibiotic resistance. The type VII secretion system [ESX-4], responsible for translocating EsxT and EsxU effectors, plays a key role in intracellular survival and bacterial pathogenesis. Legend: LAM [lipoarabinomannan], LM [lipomannan], PIM [phosphatidyl-inositol mannoside], PL [phospholipid], TDM [trehalose dimycolate], TPP [trehalose polyphleate], and TMM [trehalose monomycolate]. Figure created with BioRender.com.

**Figure 2 microorganisms-13-00447-f002:**
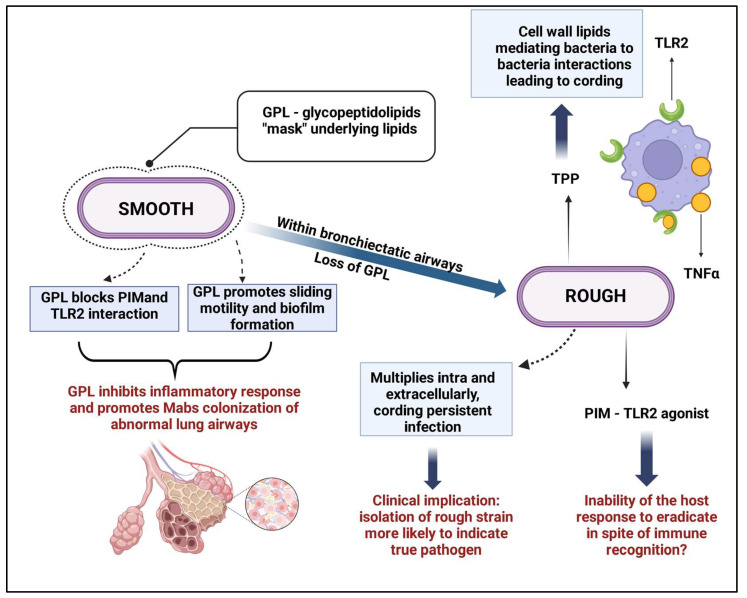
Mechanisms of immune evasion by *M. abscessus*. The smooth colony morphotype, expressing GPLs, “masks” immune recognition molecules such as PIMs and prevents the bacterial interaction of lipids like trehalose polyphleates. This action facilitates pulmonary colonization. The loss of GPLs leads to recognition by TLR2, resulting in the release of pro-inflammatory cytokines. The rough morphotype, more virulent, grows in serpentine cords, inducing macrophage apoptosis to disseminate the infection. This figure was created with BioRender.com.

**Figure 3 microorganisms-13-00447-f003:**
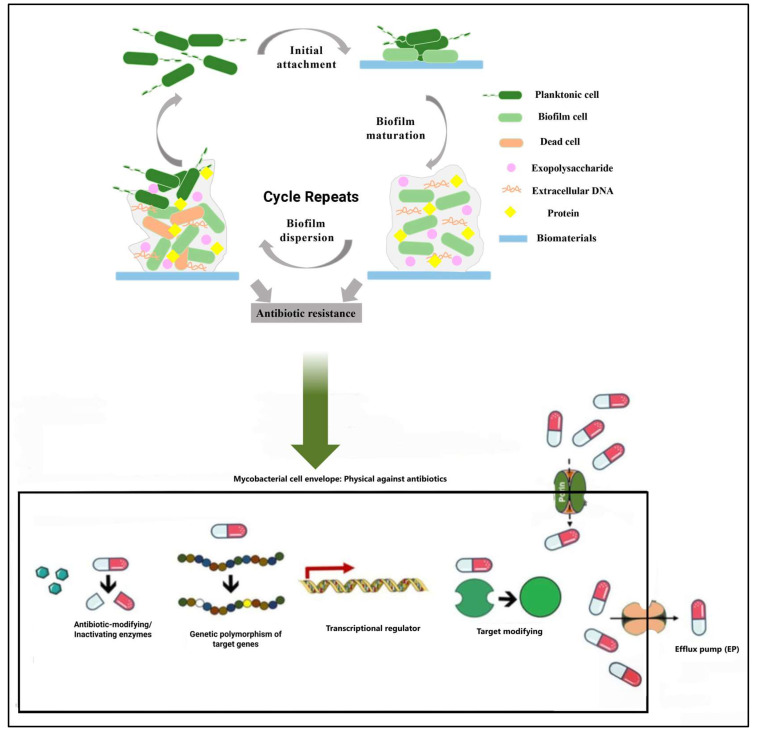
Schematic representation of *M. abscessus* biofilm formation and antibiotic resistance. Biofilm formation begins with the reversible attachment of planktonic cells [dark green] to a surface [blue]. The bacteria then form a monolayer and irreversibly adhere, producing an extracellular matrix. A microcolony forms, with multilayered growth appearing. In later stages, the biofilm matures. Finally, some cells begin to detach, and the biofilm disperses. The inherent resistance of *M. abscessus* subsp. *massiliense* is attributed to factors such as an impermeable and serous cell wall that acts as a physical [size exclusion] and chemical [hydrophobic] barrier, drug export systems, enzymes that modify or target drugs, and genetic polymorphisms in target genes. This figure was created with BioRender.com.

**Figure 4 microorganisms-13-00447-f004:**
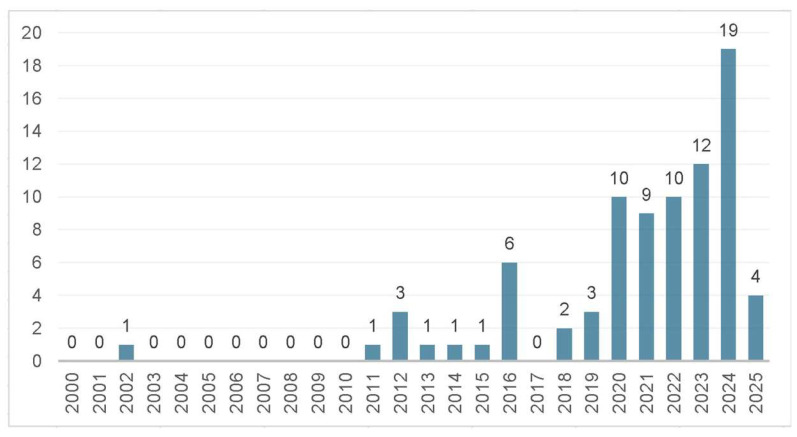
Number of annual publications from 2000 to 2025 on new therapeutic strategies for *M. abscessus* infections.

**Table 1 microorganisms-13-00447-t001:** Virulence factors of *M. abscessus* subsp. *massiliense*.

Virulence Factor	Description	Role in Virulence	References
Morphotypes	Smooth [MaSm]: rich in GPLs, forms viscous biofilms, evades the immune system. Rough [MaRg]: fewer GPLs, more invasive, forms resistant aggregates.	Adaptation to the host environment, immune evasion, resistance to adverse conditions.	[18,19]
Glycopeptidolipids [GPLs]	Abundant cell wall components in the smooth morphotype, surface lipids that affect biofilm structure and modulate the immune response.	Adhesion, biofilm formation, immune response modulation [inhibits TLR2 and TNF-α], infection persistence.	[10,18]
Trehalose dimycolate [TDM]	Glycolipid present in both morphotypes.	Contributes to virulence, particularly in biofilms formed in the lungs of cystic fibrosis patients.	[19]
Lipoarabinomannan [LAM]	Lipoglycan present in the cell wall.	Modulates the immune response.	[19,20]
Surface proteins	Adhesins, porins, etc.	Host cell adhesion, nutrient uptake, antibiotic resistance.	[19,20,21]
Biofilms	Complex structures that protect bacteria from antibiotics and the host immune system, associated with infection persistence and recurrence.	Confer antibiotic resistance, shield bacteria from the immune system, facilitate bacterial communication and infection persistence.	[19,20,21]
Extracellular matrix [ECM]	Composed of polysaccharides, proteins, eDNA, and lipids.	Provides antibiotic resistance, protects bacteria from the immune system.	[20,21]
Quorum sensing	Bacterial communication system.	Influences biofilm formation and development.	[22]

**Table 2 microorganisms-13-00447-t002:** Classes of antibiotics for the treatment of *M. abscessus* subsp. *massiliense* infections.

Antibiotic Class	Medication	Route of Administration	Recommended Dose in Adults BTS, 2017 * (90)	Observations
Macrolides	Clarithromycin	Oral	500 mg BID	Frequently used in infections caused by *M. abscessus* subsp. massiliense, which is generally sensitive to this antibiotic class [9,10].
Aminoglycosides	Amikacin	Intravenous	10–15 mg/kg QD	Commonly included in therapeutic regimens. Can also be administered via inhalation [Liposomal Amikacin for Inhalation] [3].
Beta-lactams	Cefoxitin	Intravenous	12 g daily in divided doses	Used in combination with other agents. Cefoxitin is one of the two beta-lactams commonly used to treat *M. abscessus* infections, along with imipenem [9,10].
Other	Linezolid	Oral	600 mg QD or BID	Used depending on the strain’s sensitivity profile and the patient’s tolerance [25].
	Clofazimine	Oral	50–100 mg QD	Used depending on the strain’s sensitivity profile and the patient’s tolerance [2].
	Quinolones	Oral	1 mg/kg QD	Occasionally used, depending on the strain’s sensitivity profile and the patient’s tolerance. Ciprofloxacin is a commonly used quinolone [22].

* BTS: British Thoracic Society; BID: twice daily; QD: once daily.

**Table 3 microorganisms-13-00447-t003:** New therapeutic approaches for *M. abscessus* infections.

Therapeutic Approach	Description	Observations	References
New antibiotics	Bedaquiline: Inhibits mycobacterial ATP synthase. Tedizolid: Oxazolidinone antibiotic with activity against various non-tuberculous mycobacteria. Tiopetide (NF10011): produced by *Streptomyces* sp.	Bedaquiline has been approved for the treatment of multidrug-resistant and extensively drug-resistant tuberculosis. Tedizolid has demonstrated in vitro activity against *M. abscessus*. NF1001 exhibits activity against both planktonic forms (free cells) and biofilms of NTM, showing a reduction in bacterial load.	[8,10,12,14,17,34]
Antiparasitic agent	Bithionol: This antiparasitic agent has shown significant antimicrobial activity against *M. abscessus*, including the ability to eliminate biofilms.	Bithionol’s biofilm-eradicating efficacy is notable, achieving up to 99.9% elimination of biofilm bacteria at appropriate concentrations.	[48,50]
Phage therapy	Use of bacteriophages to infect and destroy *M. abscessus* bacteria.	Phages may be used in combination with antibiotics. Case studies have reported promising results in treating *M. abscessus* infections.	[2,37,40]
Bacteriophage	Modulation of the host immune response to enhance the ability to combat *M. abscessus* infection.	Research in this area is still preliminary. Potential strategies include the use of cytokines or other immunomodulatory molecules.	[5,10,40,41]
Beta-lactamase inhibitors	Avibactam: Inhibits the beta-lactamase Bla_Mab, which is responsible for beta-lactam resistance in *M. abscessus*.	Adding avibactam to therapeutic regimens containing beta-lactams may improve their efficacy.	[23,24,25,48]
Biofilm-targeted approaches	Strategies to inhibit biofilm formation, disperse existing biofilms, or increase the susceptibility of bacteria in biofilms to antibiotics.	Tween 80: A detergent that can disaggregate *M. abscessus* biofilms, making the bacteria more susceptible to antibiotics. Oxygenation: Increasing oxygen availability can enhance the activity of some antibiotics against *M. abscessus.* Iron chelator: The bacteria have mechanisms to survive in environments with low iron levels.	[6,20,29,41,42,43,44]
Enzymes	Enzymes that degrade the components of the extracellular matrix of the *M. abscessus* biofilm.	Enzymes with degradative potential include phospholipases, carbohydrases, proteases, and DNases.	[11,34,46,47,51,52]
Genetic editing and nanotechnology	Genetic editing and nanotechnology emerge as promising tools in the fight against *M. abscessus* infections.	Genetic editing allows for the manipulation of genes essential to the bacteria’s virulence and resistance, while nanotechnology provides solutions for drug delivery, biofilm destabilization, and infection monitoring.	[51,53]

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
