# Peer review of "Mycobacterium abscessus subsp. massiliense: Biofilm Formation, Host Immune Response, and Therapeutic Strategies"

_microorganisms, 2025, doi:10.3390/microorganisms13020447_

Round 1
Reviewer 1 Report
Comments and Suggestions for Authors
This article is comprehensive and covers a wide range of topics, including the pathogenic mechanisms, biofilm formation, immune evasion, and therapeutic strategies of M. abscessus subsp. massiliense, addressing various aspects of infectious disease research. Particularly in describing the bacterium's pathogenicity and immune evasion mechanisms, the author considers the challenges in clinical treatment, highlighting the clinical application potential of this research. Additionally, the article is well-structured, with a clear framework that logically progresses from the bacterium’s biological features, pathogenic mechanisms, to treatment methods. However, there are several issues to address:
1. In the “2.1 Characteristics” section, the subheading "2.1" appears. Typically, “2.1” would be a subsection under a broader chapter, but here it appears directly after the second chapter without a corresponding chapter title (e.g., “2.2”). This structure may confuse readers and does not conform to the typical format for review articles. I suggest removing the “2.1” designation and combining the content of the second chapter into a single section with a clear title, such as “Characteristics of Mycobacterium abscessus subsp. massiliense.”
2. There is insufficient logical connection between the various sections, especially between “Virulence Factors,” “Biofilm Formation,” and “Therapeutic Strategies.” While these sections each discuss pathogenic mechanisms, biofilm formation, and treatment methods, their interconnections are not adequately emphasized. For example, while the biofilm formation mechanism is described with respect to its impact on antibiotic resistance, the discussion of therapeutic strategies does not address how to break down biofilms or enhance drug penetration. I recommend strengthening the transitions between sections, particularly in the therapeutic strategies section, by clearly indicating how these strategies are designed to address the previously discussed biofilm structure and immune evasion mechanisms. Additionally, when introducing biofilms, the article could link immune evasion mechanisms directly to treatment strategies, discussing which drugs or methods can disrupt biofilm barriers and enhance immune responses and drug efficacy.
3. The discussion of new therapeutic strategies and immune evasion mechanisms lacks depth. Although phage therapy, immunotherapy, and other emerging treatments are mentioned, their mechanisms, preclinical studies, or clinical trial progress are not thoroughly explained. Specifically, phage therapy is mentioned as promising but lacks an in-depth exploration of its application to M. abscessus infections. Furthermore, the immune evasion mechanisms are discussed in a rather general way; while the role of biofilms in immune evasion is mentioned, there is little detail on how M. abscessus modulates immune responses (such as reducing TNF-α secretion or evading macrophage killing). I suggest expanding the section on new therapeutic strategies to discuss the principles of each approach, current research progress, preclinical data, and challenges, with particular focus on how phage therapy works, how it penetrates biofilms, and how it overcomes drug resistance. Additionally, the immune evasion section should be more thorough, exploring how M. abscessus survives within the host immune system and how this impacts treatment strategies. Referencing recent literature on how M. abscessus alters immune response pathways (such as modifying TLR2 signaling) would be beneficial.
4. There is considerable repetition between the “Virulence Factors” and “Biofilm Formation” sections. For instance, the composition, function, and role of biofilms in immune evasion and drug resistance are discussed in both sections, leading to redundant information. I recommend consolidating related content to avoid repetition. In the “Virulence Factors” section, focus on discussing pathogenic factors and biofilm formation mechanisms, while in the “Biofilm Formation” section, delve more into strategies to disrupt biofilms and enhance treatment approaches, rather than repeating the biofilm components and functions.
5. While the article covers current therapeutic strategies, the discussion on innovative treatments is somewhat lacking. The strategies for treating M. abscessus appear conventional, with little exploration of cutting-edge technologies such as gene editing, vaccine development, and nanomedicine. I suggest enhancing the discussion of advanced therapies, especially those addressing M. abscessus's unique biofilm and immune evasion mechanisms. For example, you could discuss how gene editing might interfere with biofilm formation, or how nanotechnology could be used to develop drug delivery systems targeting M. abscessus, improving therapeutic efficacy and specificity.
6. To enhance readability and convey complex concepts more effectively, I recommend incorporating several figures that can help readers understand the mechanisms discussed. Suggestions include: 1) a diagram showing M. abscessus's morphological features and cell wall structure, aiding in the understanding of its pathogenicity; 2) a diagram illustrating the biofilm formation process, highlighting its relation to drug resistance and immune evasion; 3) a flowchart depicting the immune evasion mechanisms, helping to explain how the bacterium survives in the host immune response; and 4) a summary chart comparing current and emerging therapeutic strategies, focusing on their effectiveness against biofilms and immune evasion mechanisms. These visuals would help clarify key content and deepen the reader’s understanding of treatment strategies.
7. Some paragraphs contain long sentences with dense information, which may hinder reader comprehension. Particularly when describing complex treatment strategies, the sentences can be information-heavy and difficult to digest. I recommend making the language more concise and clear, especially by breaking up complex treatment strategies into shorter, easier-to-understand sections. Additionally, avoid using excessively long sentences, and instead split complex ideas into simpler statements to improve the readability and flow of the article. Below are some suggested modifications for complex sentences:
1)"Biofilms are structured communities of microorganisms that adhere to surfaces and encapsulate themselves in a self-produced extracellular matrix, which confers antimicrobial resistance and allows the microorganisms to evade the host immune response."
Suggested revision: "Biofilms are communities of microorganisms that adhere to surfaces and form a self-produced extracellular matrix. This matrix provides antimicrobial resistance and helps microorganisms evade the host immune response."
2)"Current therapies, which rely on prolonged regimens of multiple antibiotics, often prove ineffective and result in significant side effects, making it essential to explore novel therapeutic strategies that target bacterial resistance and biofilm destruction."
Suggested revision: "Current therapies, which involve prolonged use of multiple antibiotics, are often ineffective and cause significant side effects. Therefore, it is essential to explore novel strategies that target bacterial resistance and biofilm destruction."
3)"In addition to pulmonary infections, M. abscessus subsp. massiliense is also a notable causative agent of skin and soft tissue infections, often occurring after surgical procedures, trauma, or the use of medical devices, and these infections are characterized by their persistent nature and propensity for recurrence, posing considerable challenges for clinical management."
Suggested revision: "M. abscessus subsp. massiliense causes not only pulmonary infections but also skin and soft tissue infections, particularly after surgical procedures, trauma, or the use of medical devices. These infections are persistent and often recur, posing significant challenges for clinical management."
4)"While the host’s immune response is critical in controlling the infection, it is often ineffective against bacteria within biofilms, which shield them from immune cells and antimicrobial agents, making it difficult to eradicate infections."
Suggested revision: "While the host immune response is essential for controlling infections, it is often ineffective against bacteria within biofilms. These biofilms shield bacteria from immune cells and antimicrobial agents, making infections difficult to eradicate."
5)"The combination of different therapeutic approaches, such as using antibiotics in conjunction with phage therapy, may be particularly promising for overcoming drug resistance and infection persistence, which are major challenges in treating M. abscessus infections."
Suggested revision: "Combining different therapeutic approaches, such as antibiotics with phage therapy, may help overcome drug resistance and persistent infections, which are major challenges in treating M. abscessus."
Author Response
Please check the attachment, thank you.

Reviewer 2 Report
Comments and Suggestions for Authors
Overall the manuscript is well written and provides a useful review of Mycobacterium abscessus subsp. massiliense virulence and therapy. Only a few minor comments:
page 2, paragraph 5, ;line 7. The sentence reads that "biofilms express antibiotic resistance genes" which is awkward. State rather that biofilm formation can promote bacteria within the biofilm to express antibiotic resistance genes.
There is a fair amount of duplication within the manuscript. For example the description of a biofilm is duplicated on pages 2, 4, and 5. Please review to avoid duplication.
The section on Therapeutic Strategies has major redundancies. The section on pages 5 and 6 are essentially repeated on pages 6-8. Please review these sections and combine. The material is good, just duplicative.
Author Response
RESPONSE TO REVIEWER 2'S COMMENTS
Overall the manuscript is well written and provides a useful review of Mycobacterium abscessus subsp. massiliense virulence and therapy. Only a few minor comments:
page 2, paragraph 5, ;line 7. The sentence reads that "biofilms express antibiotic resistance genes" which is awkward. State rather that biofilm formation can promote bacteria within the biofilm to express antibiotic resistance genes.
Response: We appreciate your observation and have made the substitution as suggested: “biofilm formation can promote bacteria within the biofilm to express antibiotic resistance genes”(lines 66-67).
There is a fair amount of duplication within the manuscript. For example the description of a biofilm is duplicated on pages 2, 4, and 5. Please review to avoid duplication.
Response: We appreciate your observation regarding the duplication of information in the manuscript. We recognize that the description of biofilms was repeated in some sections, which could compromise the conciseness and clarity of the text. We have carefully reviewed the manuscript and eliminated the redundancies.
The section on Therapeutic Strategies has major redundancies. The section on pages 5 and 6 are essentially repeated on pages 6-8. Please review these sections and combine. The material is good, just duplicative.
Response: We appreciate your observation regarding the redundancy in the "Therapeutic Strategies" section. We agree that the repetition of information between pages 5 and 6, and between pages 6 and 8, compromises the conciseness and clarity of the text. We have carefully reviewed the section and removed the redundancies, consolidating the information on each therapeutic strategy into a single comprehensive description.

Reviewer 3 Report
Comments and Suggestions for Authors
Here are following my comments;
1: Table 1 needs to be added with the concentration of antibiotics as recommended dosages and also provide references.
2: Separate references in a new column in Table 2.
3: Provide mechanisms related to model figures for the (a) Biofilm Formation, (b) Host Immune Response, and (c) Therapeutic Strategies.
4: In the section of 6. Therapeutic Strategies>>>>Author needs to expand more. High section wise the possible strategies.
5: Provide statistical information about the therapeutic strategies reported year-wise from 2000 to 2025 (bar graph related to the publication years).
6: Make a table related to the list of virulence factors reported in the Mycobacterium.
Author Response
RESPONSE TO REVIEWER 3'S COMMENTS
Here are following my comments;
1: Table 1 needs to be added with the concentration of antibiotics as recommended dosages and also provide references.
Response: We appreciate the suggestion and have included the antibiotic concentrations with the recommended dosages in Table 2. We also added the references that support the information in the table.
2: Separate references in a new column in Table 2.
Response: We appreciate the suggestion, and we include the column.
3: Provide mechanisms related to model figures for the (a) Biofilm Formation, (b) Host Immune Response, and (c) Therapeutic Strategies.
Response: We appreciate the suggestion. Figures illustrating the mechanisms of biofilm formation (Figure 3), mechanisms of immune evasion (Figure 2), and therapeutic strategies (Table 3) have been added.
4: In the section of 6. Therapeutic Strategies>>>>Author needs to expand more. High section wise the possible strategies.
Response: We appreciate the suggestion and have added the references in a new column in Table 3. This change was made because Table 2 became Table 3 after we incorporated the previous suggestions.
5: Provide statistical information about the therapeutic strategies reported year-wise from 2000 to 2025 (bar graph related to the publication years).
Response: We appreciate the valuable suggestion to include statistical information on therapeutic strategies reported annually. In response to your request, we have created a graph illustrating the number of therapeutic strategies for Mycobacterium abscessus infections reported each year from 2000 to 2025 (figure 3). The data, compiled from searches in databases such as Scopus, Web of Science (WOS), CINAHL, EMBASE, WHO, and PubMed/Medline, reveal a significant increase in the number of reported strategies in 2024, contrasting with the lack of records in some years between 2000, 2001, and 2003 to 2010. This temporal disparity highlights the growing scientific interest in developing new approaches to combat this challenging pathogen, emphasizing the importance of continued research to expand the therapeutic arsenal against M. abscessus subsp. massiliense infections.
We believe that the inclusion of the graph enhances the manuscript by providing a clear view of the temporal evolution of research on therapeutic strategies for M. abscessus. We hope that this additional information contributes to the understanding of the current landscape and future perspectives in the treatment of this infection.
6: Make a table related to the list of virulence factors reported in the Mycobacterium.
Response: In response to the reviewer’s valuable suggestion, we have created a detailed Table 1 in Section 3 of the manuscript, which presents the main virulence factors of Mycobacterium abscessus, including cell wall components, secreted proteins, and other factors that contribute to the bacterium's pathogenicity. Each factor is concisely described, and its role in virulence is specified, with references supporting the information. We believe that the inclusion of this table significantly enhances the manuscript, making it clearer, more organized, and informative.

Round 2
Reviewer 1 Report
Comments and Suggestions for Authors
Thank you for submitting the revised version of your manuscript. I have carefully reviewed the changes, and I am pleased to see that all the key issues raised in my initial review have been addressed. The revisions have significantly improved the clarity, structure, and depth of the article, making it more comprehensive and reader-friendly.
Specifically, the consolidation of sections, enhanced logical flow between topics, and the addition of more detailed discussions on therapeutic strategies and immune evasion mechanisms have strengthened the manuscript. The inclusion of suggested figures and the refinement of language have also improved the overall readability and effectiveness of the article.
In my opinion, the manuscript is now well-polished and suitable for publication. I appreciate the effort you have put into revising the work, and I believe it will make a valuable contribution to the field.
Congratulations on your work, and I wish you continued success in your research.
Reviewer 3 Report
Comments and Suggestions for Authors
Author responded all raised questions